# Structural and Thermal Stability of CrZrON Coatings Synthesized via Reactive Magnetron Sputtering

Sung-Min Kim [1] and Sang-Yul Lee [2,*]

1   Heat and Surface Technology R&D Department, Korea Institute of Industrial Technology (KITECH), Incheon 21999, Republic of Korea; sungminkim@kitech.re.kr
2   Department of Materials Engineering, Korea Aerospace University, Goyang 10540, Republic of Korea
*   Correspondence: sylee@kau.ac.kr

**Abstract:** This research manuscript investigates the structural and thermal stability of CrZrON coatings synthesized through reactive magnetron sputtering. The coatings were deposited at different temperatures with 120 °C and 400 °C, and with varying oxygen-to-reactive gas ratios in the range of 8.3% to 25.7%. The average chemical composition, crystallographic orientation, microstructure, lattice parameter, crystallite size, and hardness of the coatings were evaluated. The results revealed that the coatings deposited at a lower temperature of 120 °C exhibited a columnar structure, while those deposited at a higher temperature of 400 °C showed a transition towards a featureless or amorphous structure. The lattice parameter and crystallite size were influenced by the deposition temperature and oxygen ratio, indicating the incorporation of oxygen into the coatings. Hardness measurements demonstrated that the coatings' hardness decreased from 33.7 GPa to 28.6 GPa for a process temperature of 120 °C and from 32.1 GPa to 25.7 GPa for 400 °C with an increase in the oxygen ratio, primarily due to the formation of oxygen-rich compounds or oxides. Additionally, annealing experiments indicated that the coatings with featureless or amorphous structures exhibited improved thermal stability, as they maintained their structural integrity without delamination even at high annealing temperatures.

**Keywords:** CrZrON; oxynitride; hardness; thermal stability

## 1. Introduction

CrN coatings have been widely used in the tool and forming industries due to their excellent mechanical properties and wear behaviors [1–3]. However, further research has been devoted to the development of ternary systems to meet the harsh criteria of a fast-changing industry. Especially in an attempt to improve the hardness, friction resistance, and oxidation-resistance of CrN, the incorporation of elements such as Zr, Al, or Si into a Cr-N coating has been actively introduced [4–14].

Among the Cr-based ternary transition-metal nitrides, CrZrN coatings provide quite unique characteristics relative to CrN [4,6,8,9]. Kim et al. [4] reported that the maximum hardness of CrZrN with an increase of up to 12.1 at.% of Zr content reached 34 GPa, as well as the tribological properties from the ball-on-disc wear test, which revealed no film failure compared to CrN. Feng et al. [8] also found that the content ratio of Zr to Cr played an important role in enhancing mechanical and tribological properties. They proposed that the increased hardness of Cr-Zr-N originated from solid solution hardening and the lattice distortion in the film. However, as reported in our previous studies, the thermal stability of the CrZrN coating significantly deteriorates at elevated temperatures up to 500 °C, resulting in a hardness and friction coefficient almost identical to those of Cr-N [6]. This phenomenon iss mostly due to the fact that the CrZrN coating was oxidized primarily by the inward transport of oxygen, along with the outward diffusion of Cr and Zr to a small extent [15]. To overcome the inadequate high-temperature applications of CrZrN coatings,

structural modification is necessary to ensure the blockage of the oxygen pathway in CrZrN coatings. One of the strategic candidates for enhancing thermal stability is modulating structure by adding an active oxygen element, i.e., the formation of oxynitride. It has been well established that oxynitride coatings have a beneficial effect on thermal stability since the formation of a protective oxide layer on the surface impedes the inward diffusion of oxygen [16,17]. Nevertheless, the oxidation resistance and thermal stability of CrZr oxynitride coatings have never been investigated in depth until now.

As the thermal stability of the CrZrN coatings is limited to less than 500 °C, aim of this study is to investigate the structural evolution and thermal stability of CrZrON coatings as a function of the oxygen-to-nitrogen flow ratio and deposition temperature during the sputtering process. The development of thermally stable CrZrON coatings by the incorporation of oxygen into CrZrN would become valuable insights for the advancement of scientific knowledge and practical applications in industries such as tools and forming.

## 2. Materials and Experimental Procedures

### 2.1. Deposition Conditions

CrZrON coatings were deposited on two types of substrates, Si (100) wafers, using a reactive magnetron sputtering system. A segment target composed of Cr and Zr (volume fraction 1:1) was used as a source material [18]. Prior to deposition, the base pressure of the sputtering chamber was pumped down to less than $2.6 \times 10^{-3}$ Pa, and pre-sputtering was carried out to clean the substrate surface at the Ar pressure of 0.4 Pa with a pulsed DC power of 0.5 kW. Subsequently, the CrZrON coatings were synthesized to be ca. 2 μm thick using a pulsed DC power of 0.7 kW (frequency: 25 kHz, duty ratio: 70%). The working pressure with the mixture of Ar of 0.4 Pa and $N_2 + O_2$ of 0.16 Pa gas was fixed at a total pressure of 0.56 Pa. To modulate the structure of coatings, the ratio of $O_2/(O_2 + N_2)$ gas pressure was varied to be 8.3%, 16.7%, and 25%, and the deposition temperature was controlled to be 120 °C and 400 °C, respectively. During deposition, the bias voltage and rotation speed of the substrate were maintained at −100 V and 10 rpm, respectively.

### 2.2. Characterization of Coatings

Chemical compositions of the coatings were acquired using energy-dispersive X-ray spectroscopy (EDX) installed on scanning electron microscopy (SEM, Tescan/VEGA II LMU, Brno, Czech Republic). The crystalline structure and phase were characterized via X-ray diffraction (XRD, Rigaku/Ultima IV, Tokyo, Japan) with Cu Kα radiation ($\lambda$ = 0.15418 nm). Based on XRD results, Scherrer equation was used to calculate the crystallite size as follows [19]:

$$D = \frac{0.9\,\lambda}{d\,\cos\theta} \tag{1}$$

where $D$ is crystallite size, $\lambda$ is wavelength ($\lambda$ = 0.15418 nm), d is FWHM (full width at half maximum intensity of the peak in radians), and θ is Bragg's diffraction angle. In addition, the lattice parameters were calculated from the interplanar spacing using the Miller indices of the crystal planes.

Hardness was measured via a microhardness-testing system (Fischerscope H100C, Sindelfingen, Germany) with a Poisson ratio of 0.3, a load of 25 mN, and 10 s of loading time by measuring a depth less than 0.2 μm to avoid a possible substrate effect. For the reliability of the hardness value, the hardness of films was measured at least 10 times.

For evaluating the thermal stability of the coatings, the annealing was performed at a temperature of 500 °C and 600 °C in air for an hour, and then they were cooled down in a furnace.

## 3. Results and Discussion

The CrZrON coatings were synthesized via the reactive magnetron sputtering process, which employed the simultaneous sputter of a segment target composed of Cr and Zr (vol. ratio of 1:1) in the presence of $O_2$ and $N_2$ reactive gases. Using this strategy, the

average chemical composition of CrZrON coatings was measured via a SEM-equipped EDX, as summarized in Table 1 and Figure 1. As shown in Figure 1a, the ratio of metallic to non-metallic elements was calculated as a function of $O_2/(O_2 + N_2)$ gas flow. The $(Cr + Zr)/(O + N)$ ratio was calculated to be <1 for all quaternary CrZrON coatings, which suggests that the metal is under stoichiometry. With increasing the $O_2$ gas with a deposition temperature of 120 °C, the $(Cr + Zr)/(O + N)$ ratio tended to decrease in the range of 0.75 to 0.62, which is lower relative to the ratio range of 0.9 to 0.7 with the deposition temperature of 400 °C. Typically, the ratio of metallic to non-metallic elements for pure oxide or high oxygen content is approximately 0.66, implying the stoichiometric formation of $(Cr_xZr_{1-x})_2(O_{1-y}N_y)_3$ (see dashed line in Figure 1a) [20,21]. Although the CrZrON coatings synthesized with a deposition temperature of 120 °C seem to have an oxygen-rich composition, as evidenced by a $(Cr + Zr)/(O + N)$ ratio similar to 0.66 with an increase in the $O_2$ gas ratio, it was found that the oxygen content and the ratio of oxygen to nitrogen in coatings are much lower than those at a deposition temperature of 400 °C (see Table 1 and Figure 1b). Furthermore, the Cr/Zr content ratio of the coatings deposited at 400 °C (*ca.* 1.1) decreased compared to deposition at 120 °C (*ca.* 2.1). This result can be explained by the fact that the reactivity of $O_2$ gas with metallic species, especially Zr, among possible reactions in the chamber, was accelerated with increasing deposition temperature. Therefore, the CrZrON coatings synthesized at low deposition temperatures consist of ternary CrZr nitride rather than CrZr oxynitride or CrZr oxide, whereas high deposition temperatures give rise to the formation of oxygen-rich compounds. Such phase formation in CrZrON coatings is comprehensively discussed in the XRD analysis.

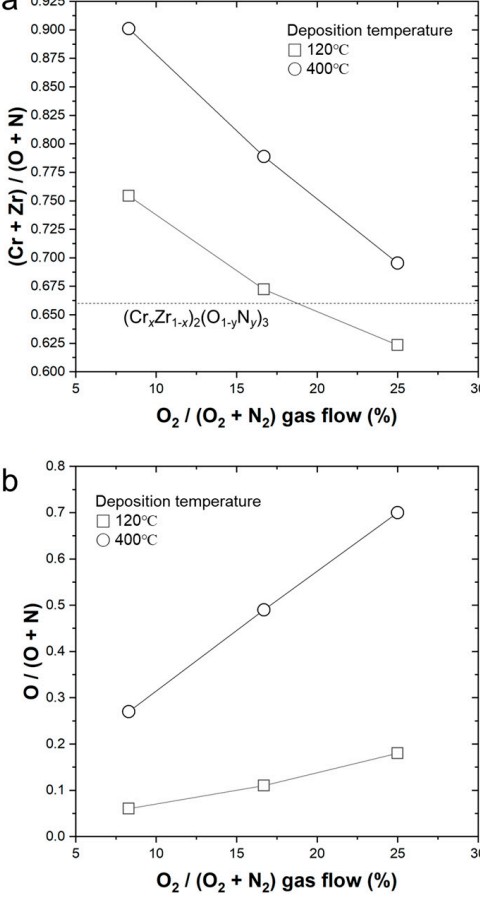

**Figure 1.** (**a**) $(Cr + Zr)/(O + N)$ ratio and (**b**) $O/(O + N)$ of CrZrON coatings corresponding to (**a**) $(Cr + Zr)/(O + N)$ ratio and (**b**) $O/(O + N)$ ratio as a function of $O_2/(O_2 + N_2)$ gas flow.

**Table 1.** Chemical composition of CrZrON coatings measured via EDS.

| Deposition Temperature (°C) | O$_2$/(O$_2$ + N$_2$) Gas Flow Ratio (%) | Chemical Composition (at.%) |
|---|---|---|
| 120 | 8.3 | Cr$_{29.0}$ Zr$_{14.0}$ O$_{3.4}$ N$_{53.6}$ |
| | 16.7 | Cr$_{27.4}$ Zr$_{12.8}$ O$_{6.7}$ N$_{53.1}$ |
| | 25.0 | Cr$_{26.5}$ Zr$_{11.9}$ O$_{11.1}$ N$_{50.5}$ |
| 400 | 8.3 | Cr$_{24.9}$ Zr$_{22.5}$ O$_{14.1}$ N$_{38.5}$ |
| | 16.7 | Cr$_{23.3}$ Zr$_{20.8}$ O$_{27.2}$ N$_{28.7}$ |
| | 25.0 | Cr$_{21.8}$ Zr$_{19.3}$ O$_{41.2}$ N$_{17.9}$ |

The XRD analysis was performed on CrZrON coatings deposited under different conditions, specifically at deposition temperatures of 120 and 400 °C, as shown in Figure 2. The influence of the O$_2$/(O$_2$ + N$_2$) gas flow ratio on the crystallographic orientation and structure of the films was investigated. At a deposition temperature of 120 °C, the XRD results revealed interesting observations. When the O$_2$/(O$_2$ + N$_2$) ratio was initially set at 8.3%, a strong CrZrN (200) peak was observed, indicating a preferential growth of the CrZrN phase. CrN and ZrN both crystallize in a rock-salt (NaCl) structure and have similar lattice parameters. This similarity allows for the formation of a solid solution between the CrN (200) and ZrN (200) planes, resulting in the CrZrN (200) phase observed in the XRD analysis. However, as the O$_2$/(O$_2$ + N$_2$) ratio increased to 25%, the intensity of the CrZrN (200) peak gradually decreased. Concurrently, the appearance of a strong (104) Cr$_2$O$_3$ peak in the XRD pattern suggested the formation of Cr$_2$O$_3$ alongside the CrZrN phase. Moreover, the (111) and (200) peaks of CrZrN were accompanied by the emergence of a strong Cr$_2$O$_3$ peak, indicating the coexistence of these phases in the deposited coating. In the case of a deposition temperature of 400 °C, the XRD results demonstrated distinct characteristics. Initially, at an O$_2$/(O$_2$ + N$_2$) ratio of 8.3%, a dominant peak corresponding to the CrZrN (200) plane was observed, indicating a preferred crystallographic orientation of the film. However, as the O$_2$/(O$_2$ + N$_2$) ratio increased to 25%, the intensity of the CrZrN (200) peak gradually decreased. Notably, at the 25% O$_2$/(O$_2$ + N$_2$) ratio, the CrZrN peaks were no longer discernible. Additionally, the sharp peaks observed for Cr$_2$O$_3$ at lower O$_2$/(O$_2$ + N$_2$) ratios were nearly diminished, and broad peaks were observed instead. These broad peaks suggest a potential transformation towards an amorphous or featureless structure in the coating. The formation of an amorphous structure at relatively high temperatures and high O$_2$ ratios could be attributed to the thermodynamic and kinetic processes occurring during the deposition of the CrZrON coatings. At high temperatures, atoms or ions possess higher kinetic energy, promoting their mobility and facilitating diffusion. This enhanced diffusion could lead to atomic rearrangements and the breakdown of long-range order, contributing to the formation of an amorphous structure [22–24]. In addition, in the presence of high oxygen content, chemical reactions between oxygen and the material's constituents can occur. These reactions may involve the formation of oxides, oxygen-rich compounds, or chemical reactions with existing crystalline phases. These chemical transformations disrupt the crystalline lattice and favor the formation of an amorphous structure by making the energy required to maintain a crystalline structure unfavorable [23,25].

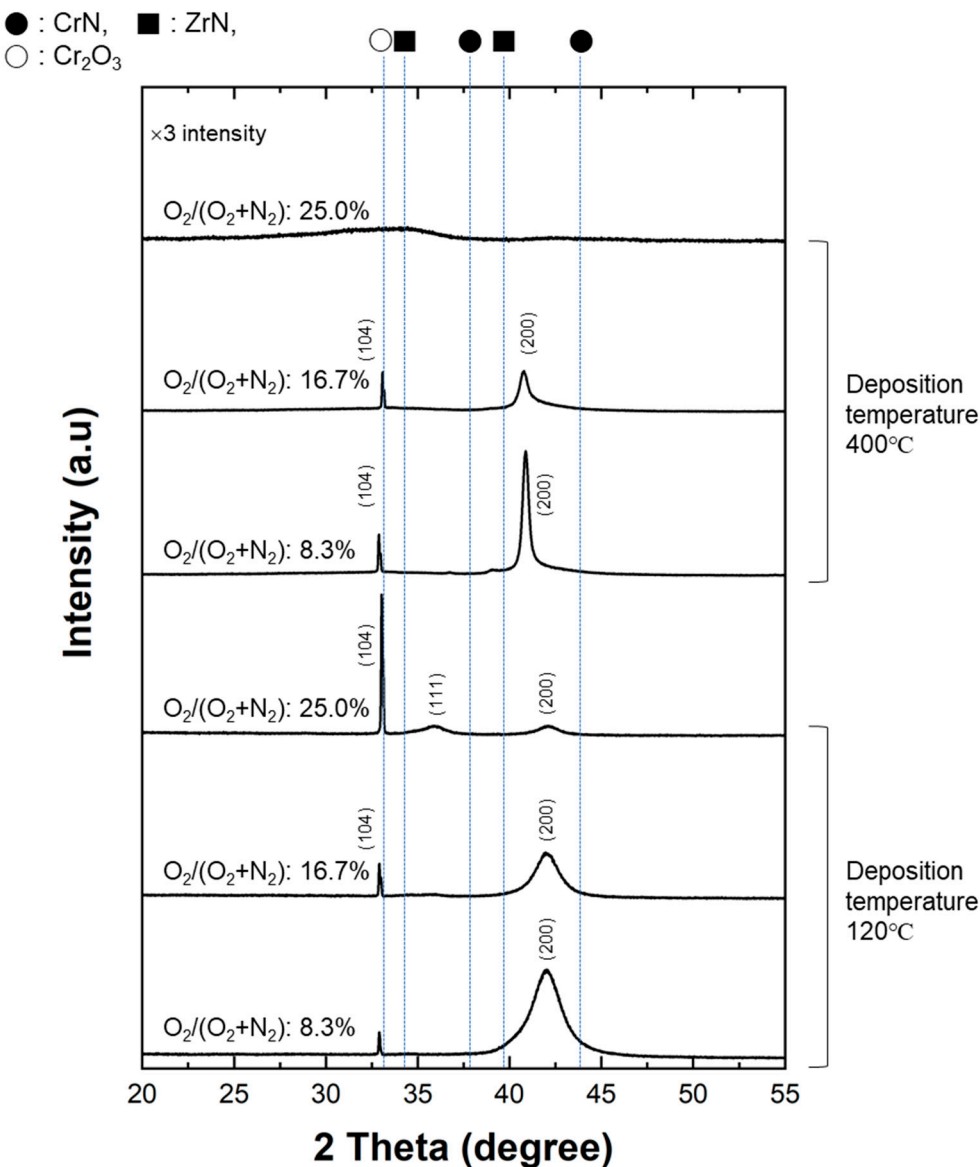

**Figure 2.** XRD of as-prepared CrZrON coatings as a function of $O_2/(O_2 + N_2)$ gas flow and deposition temperature.

The lattice parameter, crystallite size, and hardness of the CrZrON coatings were evaluated to understand the influence of the $O_2/(O_2 + N_2)$ gas flow ratio and deposition temperature on their structural and mechanical properties, as shown in Figure 3. In terms of the lattice parameter, at a deposition temperature of 120 °C, an increase in the $O_2/(O_2 + N_2)$ ratio from 8.3% to 25% led to a gradual decrease in the lattice parameter from 0.435 nm to 0.429 nm. This reduction can be attributed to the incorporation of oxygen into the coatings, resulting in the formation of oxygen-rich compounds. On the other hand, at a higher deposition temperature of 400 °C, the lattice parameter slightly decreased from 0.442 nm to 0.44 nm as the $O_2/(O_2 + N_2)$ ratio increased from 8.3% to 16.7%. However, at the highest $O_2/(O_2 + N_2)$ ratio of 25.0% and 400 °C, no clear lattice parameter was observed, suggesting the presence of a potentially amorphous or featureless structure. The crystallite size of the coatings exhibited different behaviors with respect to deposition temperature. At 120 °C, an increase in the $O_2/(O_2 + N_2)$ ratio from 8.3% to 25% resulted in a decrease in the crystallite size from 5.9 nm to 5.3 nm. This reduction can be attributed to enhanced nucleation and grain growth inhibition due to the presence of oxygen during deposition. An increase in the $O_2/(O_2 + N_2)$ ratio from 8.3% to 16.7% resulted in a decrease in the crystallite size

from 26.2 nm to 15 nm. This decrease can be attributed to the inhibition of grain growth and coalescence due to the higher concentration of oxygen in the coating. However, at the highest $O_2/(O_2 + N_2)$ ratio of 25.0% and 400 °C, no clear crystallite size was observed, indicating the potential transformation towards an amorphous or featureless structure. Regarding hardness, at 120 °C, the hardness values decreased from 33.7 GPa to 28.6 GPa as the $O_2/(O_2 + N_2)$ ratio increased from 8.3% to 25%. This decrease can be attributed to the incorporation of oxygen into the coatings, which is known to reduce the hardness of nitride films. At 400 °C, the hardness values decreased from 32.1 GPa to 25.7 GPa with an increasing $O_2/(O_2 + N_2)$ ratio from 8.3% to 25.0%. This reduction in hardness can be attributed to the combined effects of increased oxygen content. The introduction of oxygen can lead to the formation of oxygen-rich compounds, or oxides, within the coating. These compounds often have a lower hardness compared to the corresponding nitride phases. The presence of softer phases or compounds within the coating matrix can contribute to a decrease in hardness [26,27]. When oxygen is introduced into the crystal lattice, it can disrupt the arrangement of atoms and cause structural changes, leading to the formation of softer phases or compounds. The presence of these softer phases within the coating matrix can contribute to a decrease in hardness [28].

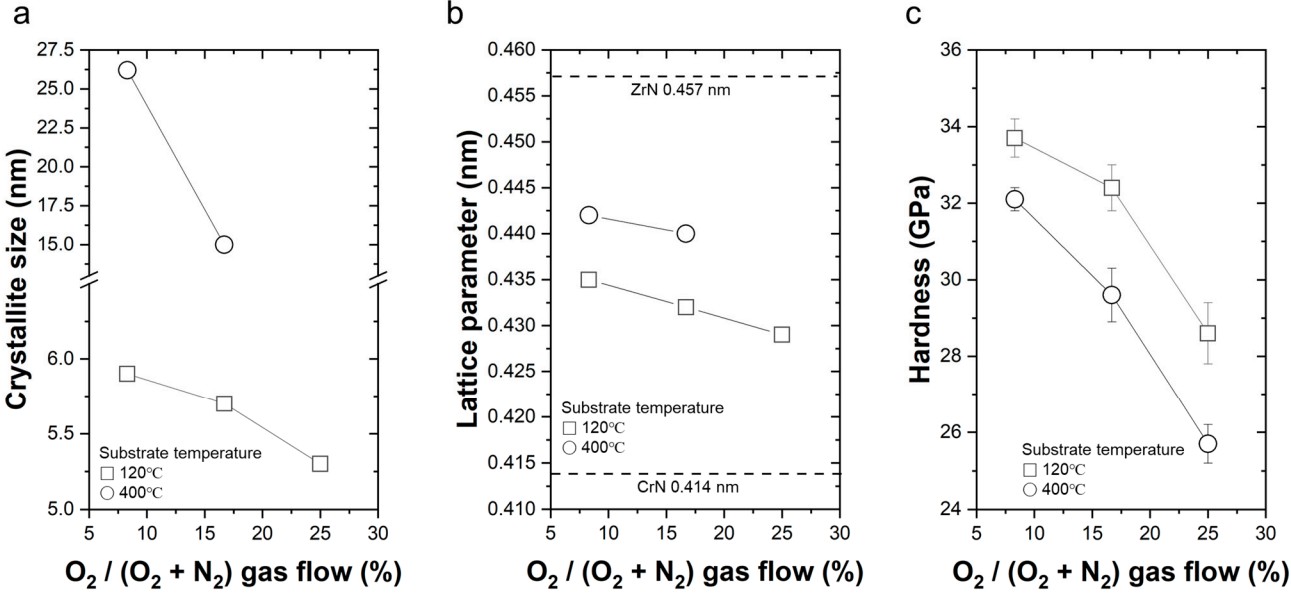

**Figure 3.** (**a**) Crystallite size, (**b**) lattice parameters, and (**c**) hardness of CrZrON as a function of $O_2/(O_2 + N_2)$ gas flow.

Figure 4 shows the microstructure of CrZrON coatings deposited under different conditions. At $O_2/(O_2 + N_2)$ in the range of 8.3% to 25.0% and lower deposition temperatures (120 °C), the coatings exhibit a columnar structure as evidenced by the FE-SEM images (Figure 4a–c). In this columnar structure, the grains of the coating align vertically, forming elongated columns. In the case of $O_2/(O_2 + N_2)$ in the range of 8.3% and 16.7% at 400 °C, the columnar structure was also taken. However, at higher oxygen ratios (25%) and higher deposition temperatures (400 °C), the FE-SEM images reveal a featureless structure. The absence of distinct columnar structure suggests a more homogeneous and possibly amorphous or nanocrystalline microstructure. The observed transition from a columnar structure to a featureless structure with an increase in oxygen ratio and deposition temperature may be attributed to the influence of oxygen on the growth kinetics and microstructure evolution during deposition. The presence of oxygen can affect the nucleation and growth processes, leading to different crystallographic orientations and grain structures.

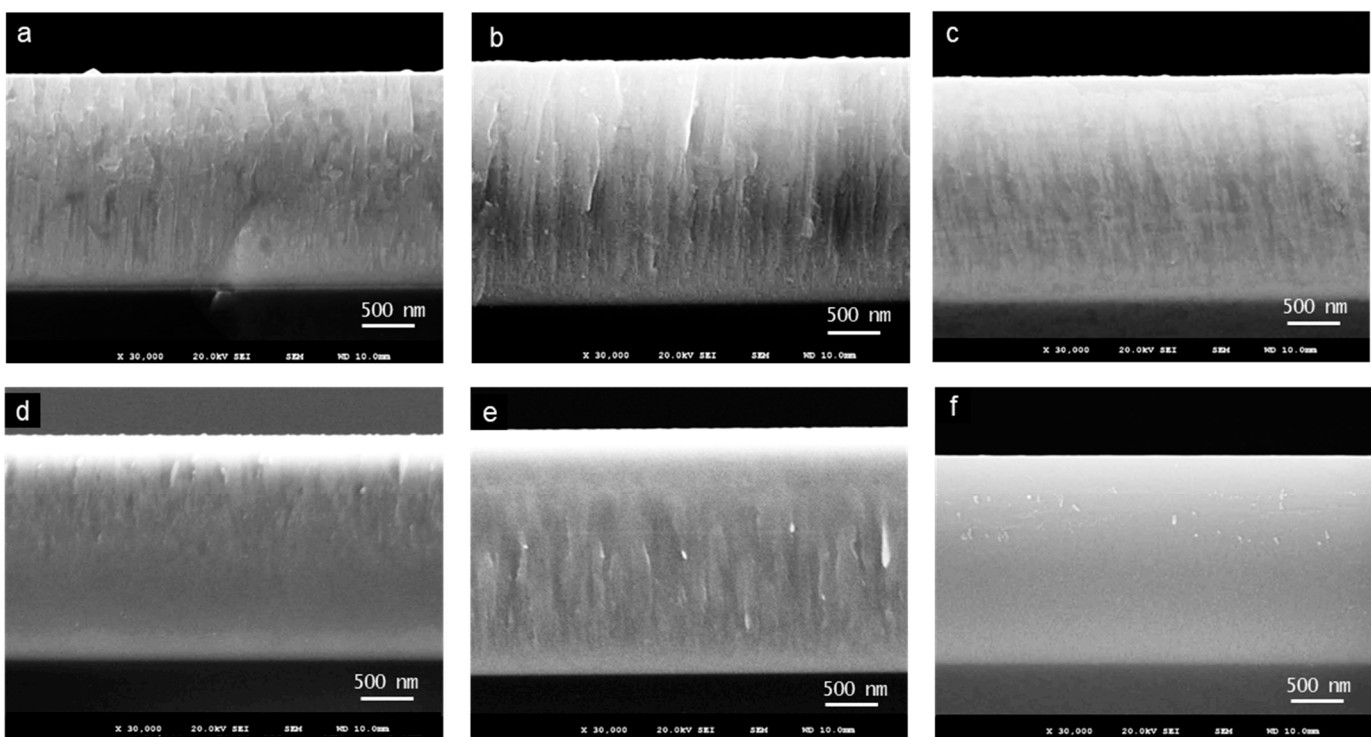

**Figure 4.** Cross-sectional FE-SEM images of CrZrON coatings corresponding to (**a**) 8.3% of $O_2/(O_2 + N_2)$ at 120 °C, (**b**) 16.7% of $O_2/(O_2 + N_2)$ at 120 °C, (**c**) 25.0% of $O_2/(O_2 + N_2)$ at 120 °C, (**d**) 8.3% of $O_2/(O_2 + N_2)$ at 400 °C, (**e**) 16.7% of $O_2/(O_2 + N_2)$ at 400 °C, and (**f**) 25.0% of $O_2/(O_2 + N_2)$ at 400 °C.

The thermal stability of the CrZrON coatings was assessed by examining their hardness under different annealing temperatures, as shown in Figure 5. For the coatings synthesized at an oxygen ratio of 8.3% and a deposition temperature of 120 °C, a high hardness of 33.7 GPa was observed at room temperature (RT). However, upon annealing at 500 °C, the hardness decreased to 9.8 GPa, indicating a reduction in the coating's strength as a result of thermal relaxation and potential grain growth. At 600 °C, delamination occurred, indicating a loss of structural integrity within the coating. Similarly, for the coatings synthesized at an $O_2/(O_2 + N_2)$ of 25% and a 120 °C deposition temperature, the hardness was 28.6 GPa at RT. After annealing at 500 °C, the hardness further decreased to 12.9 GPa, and subsequently, delamination was observed at 600 °C, indicating a critical temperature beyond which the coating's structure became unstable. In contrast, the coatings deposited at a higher temperature of 400 °C exhibited higher hardness values after annealing compared to those at 120 °C. For the 8.3% oxygen ratio, the hardness was 32.1 GPa at RT. Upon annealing at 500 °C, the hardness decreased to 11.5 GPa, and delamination also occurred at 600 °C. For the 25% oxygen ratio at 400 °C, the coating exhibited a hardness of 25.7 GPa at RT, which was relatively lower compared to the 8.3% oxygen ratio. Upon annealing at 500 °C, the hardness of the coatings with a 25% oxygen ratio decreased to 20.5 GPa. Subsequently, at 600 °C, the hardness further dropped to 18.9 GPa. Notably, while a decrease in hardness was observed, no delamination of the coatings was observed under these annealing conditions. This was due to the absence of grain boundaries and defects, which are typically present in crystalline materials and can act as initiation sites for thermal degradation processes [29,30]. In featureless or amorphous structures, the absence of long-range order and the presence of a disordered atomic arrangement result in improved resistance to grain growth, diffusion, and structural transformations at elevated temperatures [31,32]. Consequently, coatings with featureless or amorphous structures have the potential to maintain their integrity and mechanical properties even under high-temperature conditions, making them desirable for applications requiring superior thermal stability.

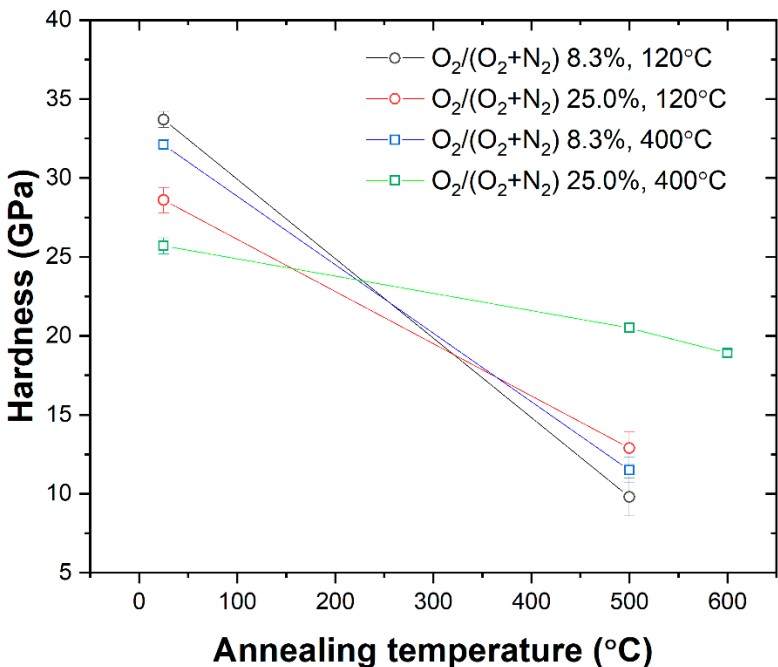

**Figure 5.** Hardness of CrZrON coatings after annealing in the range of room temperature to 600 °C.

## 4. Conclusions

In this study, the structural and thermal stability of CrZrON coatings synthesized via reactive magnetron sputtering were investigated in terms of the deposition temperature and oxygen-to-reactive gas ratio. The aim was to explore the relationship between the modified structure and the thermal stability of the coatings and to provide insights into the development of thermally stable CrZrON coatings. The main conclusions are as follows:

(1) Coatings deposited at lower temperatures exhibited a columnar structure, while those deposited at higher temperatures showed a transition towards a featureless or amorphous structure.

(2) Increasing the $O_2/(O_2 + N_2)$ ratio led to a decrease in lattice parameter, suggesting the incorporation of oxygen and the formation of oxygen-rich compounds in the coatings.

(3) Crystallite size exhibited different behaviors with respect to deposition temperature and $O_2/(O_2 + N_2)$ ratio, showing reductions due to enhanced nucleation, grain growth inhibition, and the presence of oxygen during deposition.

(4) Hardness values decreased with increasing $O_2/(O_2 + N_2)$ ratio, attributed to oxygen incorporation and the formation of oxygen-rich compounds with lower hardness compared to nitride phases.

(5) Annealing experiments demonstrated that coatings with featureless or amorphous structures exhibited superior thermal stability, maintaining their structural integrity even at high temperatures.

(6) The thermal stability of CrZrN coatings limited to 500 °C can be enhanced by incorporating oxygen, allowing CrZrON coatings to be utilized up to 600 °C, thus opening new avenues for their application in various industries.

**Author Contributions:** Conceptualization, S.-M.K.; methodology, S.-M.K.; validation, S.-M.K. and S.-Y.L.; formal analysis, S.-M.K.; investigation, S.-M.K.; resources, S.-M.K.; data curation, S.-M.K.; writing—original draft preparation, S.-M.K.; writing—review and editing, S.-Y.L.; visualization, S.-M.K.; supervision, S.-Y.L.; project administration, S.-Y.L.; funding acquisition, S.-Y.L. All authors have read and agreed to the published version of the manuscript.

**Funding:** This study was supported by the National Research Foundation of Korea (NRF) grant funded by the Korea government (MSIT) (No. 2021R1A2C1010058).

**Institutional Review Board Statement:** Not applicable.

**Informed Consent Statement:** Not applicable.

**Data Availability Statement:** Not applicable.

**Conflicts of Interest:** The authors declare no conflict of interest.

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
