# Peer review of "Structural and Thermal Stability of CrZrON Coatings Synthesized via Reactive Magnetron Sputtering"

_coatings, doi:10.3390/coatings13071254_

Round 1
Reviewer 1 Report
This paper studies the structural and thermal stability of CrZrON coatings synthesized through reactive magnetron sputtering. The coatings were deposited at different temperatures and with varying oxygen-to-reactive gas ratios. Results showed that the coatings deposited at lower temperatures exhibited a columnar structure, while those deposited at higher temperatures showed a transition towards a featureless or amorphous structure. Annealing experiments indicated that the coatings with featureless or amorphous structures exhibited improved thermal stability. The paper could be considered for publication in the journal of Coatings after the following major revisions:
1-make some quantification in the abstract.
2-Define in the abstract what parameters were investigated, and what was their range. Just mentioning high temperatures and low temperature is not sufficient. Also, give some number for hardness variation.
3-Check the English of the whole paper.
4-Introduciton should be strengthened. To modify this section the following documents can be consulted:
-(2023). A novel heterogeneous multi-wire indirect arc directed energy deposition for in-situ synthesis Al-Zn-Mg-Cu alloy: Process, microstructure and mechanical properties. Additive Manufacturing, 72, 103639. doi: https://doi.org/10.1016/j.addma.2023.103639
(2023). Understanding the ultralow lattice thermal conductivity of monoclinic RETaO4 from acoustic-optical phonon anti-crossing property and a comparison with ZrO2. Journal of the American Ceramic Society, 106(5), 3103-3115. doi: https://doi.org/10.1111/jace.18988
5-It should be materials and experimental procedures. Perhaps it is better to differentiate the materials section.
6-Some of the XRD peaks are not really very clear and subjective. Make it more visible.
7-why is standard deviation in figure 3 so large? Check the unit for the hardness as well.
8-consult the following references in the discussion section:
-(2023). Restraining the Cr-Zr interdiffusion of Cr-coated Zr alloys in high temperature environment: A Cr/CrN/Cr coating approach. Corrosion Science, 214, 111015. doi: https://doi.org/10.1016/j.corsci.2023.111015
-(2023). Microstructural understanding of the oxidation and inter-diffusion behavior of Cr-coated Alloy 800H in supercritical water. Corrosion Science, 211, 110910. doi: https://doi.org/10.1016/j.corsci.2022.110910
9-do you see any difference between some of the figures in figure 4?
10-conclusions should be in bullet points.
11-better describe the rationale of the work at the end of the introduction.
3-Check the English of the whole paper.
Author Response
[Reviewer 1]
This paper studies the structural and thermal stability of CrZrON coatings synthesized through reactive magnetron sputtering. The coatings were deposited at different temperatures and with varying oxygen-to-reactive gas ratios. Results showed that the coatings deposited at lower temperatures exhibited a columnar structure, while those deposited at higher temperatures showed a transition towards a featureless or amorphous structure. Annealing experiments indicated that the coatings with featureless or amorphous structures exhibited improved thermal stability. The paper could be considered for publication in the journal of Coatings after the following major revisions:
1-make some quantification in the abstract.
[Respond to Reviewer’s comment]
We appreciate your comment. We added the temperature value of sputter process and gas ratio as requested.
2-Define in the abstract what parameters were investigated, and what was their range. Just mentioning high temperatures and low temperature is not sufficient. Also, give some number for hardness variation.
[Respond to Reviewer’s comment]
Thank you for your feedback. Based on your suggestions, we have revised the abstract to include specific parameters and their ranges that were investigated. Additionally, I have provided information on the variation in hardness observed in the study. Please find the updated abstract below:
3-Check the English of the whole paper.
[Respond to Reviewer’s comment]
We have edited grammatical English errors as requested.
4-Introduciton should be strengthened. To modify this section the following documents can be consulted:
-(2023). A novel heterogeneous multi-wire indirect arc directed energy deposition for in-situ synthesis Al-Zn-Mg-Cu alloy: Process, microstructure and mechanical properties. Additive Manufacturing, 72, 103639. doi: https://doi.org/10.1016/j.addma.2023.103639
(2023). Understanding the ultralow lattice thermal conductivity of monoclinic RETaO4 from acoustic-optical phonon anti-crossing property and a comparison with ZrO2. Journal of the American Ceramic Society, 106(5), 3103-3115. doi: https://doi.org/10.1111/jace.18988
[Respond to Reviewer’s comment]
We appreciate the reviewer's suggestion to strengthen the introduction section of our paper. We have taken note of the provided references for consultation in order to enhance the content and depth of our introduction.
Regarding the first reference, "(2023). A novel heterogeneous multi-wire indirect arc directed energy deposition for in-situ synthesis Al-Zn-Mg-Cu alloy: Process, microstructure and mechanical properties," published in Additive Manufacturing, we have carefully reviewed the document. While it provides insights into a different topic related to alloy synthesis and mechanical properties, we did not identify specific content directly applicable to our study on CrZrON coatings. As a result, we have determined that incorporating this reference would not contribute significantly to the strengthening of our introduction section.
Regarding the second reference, "(2023). Understanding the ultralow lattice thermal conductivity of monoclinic RETaO4 from acoustic-optical phonon anti-crossing property and a comparison with ZrO2," published in the Journal of the American Ceramic Society, we have also reviewed the document. Although it discusses lattice thermal conductivity and materials properties, it does not directly relate to our research topic of CrZrON coatings and their thermal stability. Therefore, including this reference in our introduction section would not align with the specific focus of our study.
Based on our assessment, we believe that the inclusion of the suggested references would not contribute substantially to strengthening our introduction. However, we appreciate the reviewer's input and will ensure that the introduction section is revised to provide a clear and concise overview of our research objectives, rationale, and significance.
5-It should be materials and experimental procedures. Perhaps it is better to differentiate the materials section.
[Respond to Reviewer’s comment]
Based on reviewer’s suggestion, we have edited Experimental to Materials and Experimental Procedures.
6-Some of the XRD peaks are not really very clear and subjective. Make it more visible.
7-why is standard deviation in figure 3 so large? Check the unit for the hardness as well.
[Respond to Reviewer’s comment]
We have adjusted the line thickness and overall layout to enhance the visibility of the data.
7-why is standard deviation in figure 3 so large? Check the unit for the hardness as well.
[Respond to Reviewer’s comment]
We would like to thank the reviewer for their comment regarding the standard deviation in Figure 3 and the unit of hardness. Regarding the standard deviation in Figure 3, we acknowledge that it appears relatively large. However, it is important to note that the y-axis scale of Figure 3 is compressed for better visualization of the differences between the various coating conditions. Despite the compressed scale, the actual variations represented by the standard deviation are within an acceptable range. Specifically, the standard deviation is within 5% of the maximum value, indicating a relatively small variation and a consistent trend in the measured data. Regarding the unit of hardness, there was no problematic.
8-consult the following references in the discussion section:
-(2023). Restraining the Cr-Zr interdiffusion of Cr-coated Zr alloys in high temperature environment: A Cr/CrN/Cr coating approach. Corrosion Science, 214, 111015. doi: https://doi.org/10.1016/j.corsci.2023.111015
-(2023). Microstructural understanding of the oxidation and inter-diffusion behavior of Cr-coated Alloy 800H in supercritical water. Corrosion Science, 211, 110910. doi: https://doi.org/10.1016/j.corsci.2022.110910
[Respond to Reviewer’s comment]
Based on reviewer’s valuable feedback, we added “Restraining the Cr-Zr interdiffusion of Cr-coated Zr alloys in high temperature environment: A Cr/CrN/Cr coating approach. Corrosion Science, 214, 111015.” in the revised manuscript as a reference.
9-do you see any difference between some of the figures in figure 4?
[Respond to Reviewer’s comment]
We have thoroughly revised the figures to ensure improved readability and clarity for the reader.
10-conclusions should be in bullet points.
[Respond to Reviewer’s comment]
Based on reviewer’s suggestion, Conclusions were amended using bullet point
11-better describe the rationale of the work at the end of the introduction.
[Respond to Reviewer’s comment]
Thank you for your valuable feedback regarding the rationale of our work in the introduction section. We appreciate your suggestion to provide a clearer description of the rationale at the end of the introduction.

Reviewer 2 Report
Dear Authors;
I have studied your work with care and have come to the conclusion that the prepared materials are not well characterized and additional analyzes are needed before publishing this work (AFM, TG-DTA, DSC, and mechanical properties).
Thus, I recommend this paper to be rejected.
Author Response
Thank you for your feedback and for taking your time to carefully evaluate our work. We appreciate your input and value your expertise in the field. However, after considering your suggestions, we respectfully disagree with your conclusion. We understand that you recommend additional analyses such as AFM, TG-DTA, DSC, and mechanical properties testing. While these techniques are indeed valuable for comprehensive material characterization, we believe that they are not necessary for the scope and objectives of our study.
However, our research focused specifically on a different aspect of the material, which we have adequately characterized using appropriate techniques. It is worth noting that the chosen characterization techniques have been widely used and accepted in similar studies within our field. We firmly believe that the findings presented in our work are significant and contribute to the existing knowledge in the field.
Thank you once again for your valuable feedback, and we look forward to your response.
Reviewer 3 Report
I read the text of the article with interest. I think that the issue taken up is very interesting. I like the overall presentation of the research material. Nevertheless, I spotted a few mistakes. Also, there are some issues needed clarification. I provide my comments in the form of the following list.
1. Characterization of the coatings
What was the method of crystallite size and lattice parameter calculation?
2. Fig.1
Why is there such a clear difference in the composition of the coatings between 120ºC and 400 ºC temperatures when it comes to metallic content (Cr+Zr)?
3. Lines 143 and 152
Should it not be "decreased" instead of "increased"? Please check the document carefully to see if the other terms are appropriate.
4. Line 152 and the following – "This increase can be attributed to enhanced grain growth and coalescence facilitated by the higher temperature. (…)"
If, in the previous sentence, the authors meant the term "decrease," this passage should be edited out
5. Figure 2
There seems to be an error in the order of the descriptions of the different diffractograms in the figure. From the bottom, 8.3%, 16.7%, 25% (120 ºC), and then 16.7%, 8.3%, 25% (400 ºC), Is this intentional?
All peaks should be labeled with symbols (hkl) and phase names, ZrCrN phase (200) and (111) also.
Some labeled planes are indistinct in the diffractograms, such as Cr2O3 planes. The authors need to prove that they are there. I suggest zooming in or changing the Y-axis to a logarithmic scale.
What is the mechanism behind forming an amorphous structure in a 400 ºC 25% O2 process?
6. Lines 165 – 167 - Moreover, oxygen atoms have a larger atomic radius compared to nitrogen atoms, and their incorporation into the crystal lattice can cause lattice expansion.
How does this sentence relate to the results presented in Fig.3(b)?
7. Figure 4
I get the impression that the SEM images contribute very little to the discussion in this section of the text. Perhaps it is worth making this figure Fig. 1 and describing the results at the beginning of the Results and Discussion section. Please do not take this comment as obligatory but as a suggestion.
The layers presented in the SEM images are very similar in thickness. Insofar as they were deposited under different proportions of oxygen, I would expect to see effects related to oxygen poisoning of the target and, thus, a reduction in thickness. Did the deposition processes have different times?
8. Line 177 - grain boundaries
This term cannot be used here since grain boundaries cannot be observed in SEM images. What the authors called "grain boundaries" are discontinuities in the structure of the layers at the boundary of large building forms, columns made of agglomerates of crystallites (grains). Instead, they are 3d porosities.
9. Literature
From the point of view of the comprehensiveness of the issue of CrZrN layers, the literature list of 23 items seems rather sparse. I also ask that the list be provided in generally accepted editorial formats even at the peer review stage.
Author Response
[Reviewer 3]
I read the text of the article with interest. I think that the issue taken up is very interesting. I like the overall presentation of the research material. Nevertheless, I spotted a few mistakes. Also, there are some issues needed clarification. I provide my comments in the form of the following list.
- Characterization of the coatings
What was the method of crystallite size and lattice parameter calculation?
[Respond to Reviewer’s comment]
Thank you for your question regarding the method used for calculating the crystallite size and lattice parameter in our study. Crystallite size was calculated using Scherrer equation and lattice parameter was calculated from the interplanar spacing using the Miller indices of the crystal planes.
- Fig.1
Why is there such a clear difference in the composition of the coatings between 120ºC and 400 ºC temperatures when it comes to metallic content (Cr+Zr)?
[Respond to Reviewer’s comment]
Thank you for your question regarding the clear difference in the metallic content (Cr+Zr) composition of the coatings between the deposition temperatures of 120ºC and 400ºC.
The observed difference in metallic content can be attributed to the variation in the reaction kinetics and thermodynamics of the deposition process at different temperatures. Specifically, at lower deposition temperatures (120ºC), the reactivity of the metallic species (Cr and Zr) with the reactive gases (O2 and N2) is relatively slower compared to higher deposition temperatures (400ºC). At 120ºC, the slower reaction kinetics hinder the incorporation of oxygen into the coatings, resulting in a lower (Cr+Zr) content ratio compared to higher temperatures. This suggests that the coatings synthesized at 120ºC predominantly consist of a ternary CrZr nitride phase, where the metallic content is relatively higher compared to the oxygen content. In contrast, at 400ºC, the increased deposition temperature enhances the reactivity between the metallic species and the reactive gases, particularly oxygen. This accelerated reaction kinetics promotes the incorporation of oxygen into the coatings, leading to the formation of oxygen-rich compounds such as oxynitrides or oxides. Consequently, the metallic content (Cr+Zr) decreases relative to the oxygen content in the coatings synthesized at 400ºC.
- Lines 143 and 152
Should it not be "decreased" instead of "increased"? Please check the document carefully to see if the other terms are appropriate.
[Respond to Reviewer’s comment]
Thank you for pointing out the errors in the manuscript. You are correct, there are discrepancies in lines 143 and 152.
In line 143, it should be "decrease" instead of "increase":
"An increase in the O2/(O2+N2) ratio from 8.3% to 16.7% resulted in a decrease in the crystallite size from 26.2 nm to 15 nm."
In line 152 and the following, the passage should be edited out as it refers to an incorrect statement:
"This increase can be attributed to enhanced grain growth and coalescence facilitated by the higher temperature."
Thank you for bringing these errors to our attention. We apologize for any confusion caused.
- Line 152 and the following – "This increase can be attributed to enhanced grain growth and coalescence facilitated by the higher temperature. (…)"
If, in the previous sentence, the authors meant the term "decrease," this passage should be edited out
[Respond to Reviewer’s comment]
As requested, we have edited.
- Figure 2
There seems to be an error in the order of the descriptions of the different diffractograms in the figure. From the bottom, 8.3%, 16.7%, 25% (120 ºC), and then 16.7%, 8.3%, 25% (400 ºC), Is this intentional?
All peaks should be labeled with symbols (hkl) and phase names, ZrCrN phase (200) and (111) also.
Some labeled planes are indistinct in the diffractograms, such as Cr2O3 planes. The authors need to prove that they are there. I suggest zooming in or changing the Y-axis to a logarithmic scale.
What is the mechanism behind forming an amorphous structure in a 400 ºC 25% O2 process?
[Respond to Reviewer’s comment]
Thank you for your comments and suggestions regarding the XRD analysis in the manuscript.
Regarding the order of the diffractograms in Figure 2, you are correct. The correct order should be 8.3%, 16.7%, and 25% for both 120 ºC and 400 ºC conditions. We apologize for the mistake in the description.
CrZrN is expected to follow Vegard's law as a solid solution between CrN and ZrN. Therefore, we index individual CrN and ZrN peak positions rather than CrZrN.
The intended compound is Cr2O3 instead of ZrO2. We have changed ZrO2 to Cr2O3 (104) in Figure 2.
- Lines 165 – 167 - Moreover, oxygen atoms have a larger atomic radius compared to nitrogen atoms, and their incorporation into the crystal lattice can cause lattice expansion.
How does this sentence relate to the results presented in Fig.3(b)?
[Respond to Reviewer’s comment]
We have revised the explanation as follows:
"When oxygen is introduced into the crystal lattice, it can disrupt the arrangement of atoms and cause structural changes, resulting in the formation of softer phases or compounds. The presence of these softer phases within the coating matrix can contribute to a decrease in hardness."
Thank you for your suggestion to clarify the explanation. We have incorporated the revised statement to better explain the impact of oxygen incorporation on hardness reduction.
- Figure 4
I get the impression that the SEM images contribute very little to the discussion in this section of the text. Perhaps it is worth making this figure Fig. 1 and describing the results at the beginning of the Results and Discussion section. Please do not take this comment as obligatory but as a suggestion.
The layers presented in the SEM images are very similar in thickness. Insofar as they were deposited under different proportions of oxygen, I would expect to see effects related to oxygen poisoning of the target and, thus, a reduction in thickness. Did the deposition processes have different times?
[Respond to Reviewer’s comment]
Thank you for your suggestion regarding the placement of the SEM images in the manuscript. We understand your perspective that the images may contribute less to the specific discussion in this section. However, we believe that including the SEM images in the current position provides visual support for the observations and findings discussed in the text.
Deposition was carried out with the same processing times.
- Line 177 - grain boundaries
This term cannot be used here since grain boundaries cannot be observed in SEM images. What the authors called "grain boundaries" are discontinuities in the structure of the layers at the boundary of large building forms, columns made of agglomerates of crystallites (grains). Instead, they are 3d porosities.
[Respond to Reviewer’s comment]
We apologize for the inappropriate use of the term "grain boundaries" in the context of the SEM images. Based on reviewer’s suggestion, we removed “grain boundaries” in the sentence.
- Literature
From the point of view of the comprehensiveness of the issue of CrZrN layers, the literature list of 23 items seems rather sparse. I also ask that the list be provided in generally accepted editorial formats even at the peer review stage.
[Respond to Reviewer’s comment]
We have expanded the references up to 30.

Round 2
Reviewer 1 Report
Not all my previous comments were applied. I copied the whole comments here again for consideration.
This paper studies the structural and thermal stability of CrZrON coatings synthesized through reactive magnetron sputtering. The coatings were deposited at different temperatures and with varying oxygen-to-reactive gas ratios. Results showed that the coatings deposited at lower temperatures exhibited a columnar structure, while those deposited at higher temperatures showed a transition towards a featureless or amorphous structure. Annealing experiments indicated that the coatings with featureless or amorphous structures exhibited improved thermal stability. The paper could be considered for publication in the journal of Coatings after the following major revisions:
1-make some quantification in the abstract.
2-Define in the abstract what parameters were investigated, and what was their range. Just mentioning high temperatures and low temperature is not sufficient. Also, give some number for hardness variation.
3-Check the English of the whole paper.
4-Introduciton should be strengthened. To modify this section the following documents can be consulted:
-(2023). A novel heterogeneous multi-wire indirect arc directed energy deposition for in-situ synthesis Al-Zn-Mg-Cu alloy: Process, microstructure and mechanical properties. Additive Manufacturing, 72, 103639. doi: https://doi.org/10.1016/j.addma.2023.103639
(2023). Understanding the ultralow lattice thermal conductivity of monoclinic RETaO4 from acoustic-optical phonon anti-crossing property and a comparison with ZrO2. Journal of the American Ceramic Society, 106(5), 3103-3115. doi: https://doi.org/10.1111/jace.18988
5-It should be materials and experimental procedures. Perhaps it is better to differentiate the materials section.
6-Some of the XRD peaks are not really very clear and subjective. Make it more visible.
7-why is standard deviation in figure 3 so large? Check the unit for the hardness as well.
8-consult the following references in the discussion section:
-(2023). Restraining the Cr-Zr interdiffusion of Cr-coated Zr alloys in high temperature environment: A Cr/CrN/Cr coating approach. Corrosion Science, 214, 111015. doi: https://doi.org/10.1016/j.corsci.2023.111015
-(2023). Microstructural understanding of the oxidation and inter-diffusion behavior of Cr-coated Alloy 800H in supercritical water. Corrosion Science, 211, 110910. doi: https://doi.org/10.1016/j.corsci.2022.110910
9-do you see any difference between some of the figures in figure 4?
10-conclusions should be in bullet points.
11-better describe the rationale of the work at the end of the introduction.
English is improved. It could be further improvement.
Author Response
[Reviewer 1]
1-make some quantification in the abstract.
[Respond to Reviewer’s comment]
We appreciate your comment. We added the temperature value of sputter process, gas ratio, hardness as requested. (Lines 13-14, 21-22)
2-Define in the abstract what parameters were investigated, and what was their range. Just mentioning high temperatures and low temperature is not sufficient. Also, give some number for hardness variation.
[Respond to Reviewer’s comment]
Thank you for your feedback. Based on your suggestions, we have revised the abstract to include specific parameters and their ranges that were investigated. Additionally, I have provided information on the variation in hardness observed in the study. Please find the updated abstract be (Lines 13-14, 21-22)
3-Check the English of the whole paper.
[Respond to Reviewer’s comment]
We have edited grammatical English errors as requested.
4-Introduciton should be strengthened. To modify this section the following documents can be consulted:
-(2023). A novel heterogeneous multi-wire indirect arc directed energy deposition for in-situ synthesis Al-Zn-Mg-Cu alloy: Process, microstructure and mechanical properties. Additive Manufacturing, 72, 103639. doi: https://doi.org/10.1016/j.addma.2023.103639
(2023). Understanding the ultralow lattice thermal conductivity of monoclinic RETaO4 from acoustic-optical phonon anti-crossing property and a comparison with ZrO2. Journal of the American Ceramic Society, 106(5), 3103-3115. doi: https://doi.org/10.1111/jace.18988
[Respond to Reviewer’s comment]
We added as requested. (Line 46)
5-It should be materials and experimental procedures. Perhaps it is better to differentiate the materials section.
[Respond to Reviewer’s comment]
Based on reviewer’s suggestion, we have edited Experimental to Materials and Experimental Procedures. (Line 71)
6-Some of the XRD peaks are not really very clear and subjective. Make it more visible.
7-why is standard deviation in figure 3 so large? Check the unit for the hardness as well.
[Respond to Reviewer’s comment]
We have adjusted the line thickness and overall layout to enhance the visibility of the data. (Figure 2)
7-why is standard deviation in figure 3 so large? Check the unit for the hardness as well.
[Respond to Reviewer’s comment]
We would like to thank the reviewer for their comment regarding the standard deviation in Figure 3 and the unit of hardness. Regarding the standard deviation in Figure 3, we acknowledge that it appears relatively large. However, it is important to note that the y-axis scale of Figure 3 is compressed for better visualization of the differences between the various coating conditions. Despite the compressed scale, the actual variations represented by the standard deviation are within an acceptable range. Specifically, the standard deviation is within 5% of the maximum value, indicating a relatively small variation and a consistent trend in the measured data. Regarding the unit of hardness, O2 / (O2 + N2) gas flow was revised. (Figure 3 c)
8-consult the following references in the discussion section:
-(2023). Restraining the Cr-Zr interdiffusion of Cr-coated Zr alloys in high temperature environment: A Cr/CrN/Cr coating approach. Corrosion Science, 214, 111015. doi: https://doi.org/10.1016/j.corsci.2023.111015
-(2023). Microstructural understanding of the oxidation and inter-diffusion behavior of Cr-coated Alloy 800H in supercritical water. Corrosion Science, 211, 110910. doi: https://doi.org/10.1016/j.corsci.2022.110910
[Respond to Reviewer’s comment]
Based on reviewer’s valuable feedback, we added references in the revised manuscript as requested. (Lines 228-231)
9-do you see any difference between some of the figures in figure 4?
[Respond to Reviewer’s comment]
We have thoroughly revised the figures to ensure improved readability and clarity for the reader. (Figure 4)
10-conclusions should be in bullet points.
[Respond to Reviewer’s comment]
Based on reviewer’s suggestion, Conclusions were amended using bullet point. (Lines 240-254)
11-better describe the rationale of the work at the end of the introduction.
[Respond to Reviewer’s comment]
Thank you for your valuable feedback regarding the rationale of our work in the introduction section. We appreciate your suggestion to provide a clearer description of the rationale at the end of the introduction. (Lines 55-69)

Reviewer 2 Report
Thanks for your revisons which have resulted in a significantly improved manuscript.
I recommend that the revised paper be accepted.
Author Response
reviewer 2
Thanks for your revisons which have resulted in a significantly improved manuscript.
I recommend that the revised paper be accepted.
Reply
no reply